# Multi-Strategy Deployment-Time Learning and Adaptation for Navigation under Uncertainty

**Abhishek Paudel, Xuesu Xiao, and Gregory J. Stein**
Department of Computer Science
George Mason University
{apaudel4, xiao, gjstein}@gmu.edu

**Abstract:** We present an approach for performant point-goal navigation in unfamiliar partially-mapped environments. When deployed, our robot runs multiple strategies for deployment-time learning and visual domain adaptation in parallel and quickly selects the best-performing among them. Choosing between policies as they are learned or adapted between navigation trials requires continually updating estimates of their performance as they evolve. Leveraging recent work in model-based learning-informed planning under uncertainty, we determine lower bounds on the would-be performance of newly-updated policies on old trials without needing to re-deploy them. This information constrains and accelerates bandit-like policy selection, affording quick selection of the best-performing strategy shortly after it would start to yield good performance. We validate the effectiveness of our approach in simulated maze-like environments, showing improved navigation cost and cumulative regret versus existing baselines.

**Keywords:** policy selection, domain adaptation, navigation under uncertainty

## 1 Introduction

Consider a robot deployed in an unknown environment and tasked to reach an unseen point goal. The robot leverages learning to make decisions about where to go to most quickly find the goal. However, depending upon how different the training and deployment environments are, learning may not always inform good behavior and the robot may demonstrate poor performance during deployment. Thus, to perform well in a wide variety of environments, the robot must improve its behavior *during deployment*, via strategies such as visual domain adaptation or online policy learning. Problematically, there is no *one-size-fits-all* approach to deployment-time learning or adaptation: each such strategy is typically only suitable for addressing a subset of the types of changes a robot may encounter during deployment or are slow to converge, risking poor performance during a potentially lengthy or error-prone learning phase and limiting most such approaches in general.

If robots in this goal-directed navigation scenario are to perform well when deployed in arbitrary unfamiliar environments, they could ideally run many strategies for adaptation and online learning *in parallel* and rely on *run-time monitoring* to select the best-performing policy or strategy. Bandit algorithms [1, 2, 3] are potentially useful, yet are prohibitively slow to converge in this setting. Other existing work in the space of run-time monitoring [4, 5, 6, 7, 8] or policy selection for reinforcement learning [9, 10, 11, 12] are similarly limited to short time horizons or fully-known environments and are not straightforwardly applicable. Moreover, such strategies typically presume that the policies are static, posing a risk when the policies improve during deployment: deciding to *not select* a policy that was deemed to perform poorly early on—yet has improved considerably since then—results in poor overall performance. Instead, we require an approach that can reliably select between a family of continually changing (*non-stationary*) policies being learned or adapted during deployment.

8th Conference on Robot Learning (CoRL 2024), Munich, Germany.

**Goal**: Select the best of an ensemble of learning-informed visual navigation policies, even as many of those evolve during deployment via online learning or visual domain adaptation.

Performance of $\pi_{\text{ADAPT}}$ greatly improves after a round of visual domain adaptation at Trial 10:

We use deployment-time data to "replay" old trials, updating otherwise-outdated performance estimates:

Even if provided privileged information about policy performance over time, selection via a rolling average fails to quickly switch to $\pi_{\text{ADAPT}}$.

Our approach affords deploying multiple policies or strategies for online learning or domain adaptation *in parallel* and quickly choosing the best performing among them during deployment.

Figure 1: Overview of our approach for fast selection between non-stationary policies.

In this work, we present an approach for data-efficient and reliable policy selection capable of choosing the best-performing policies from an ensemble of policies, even when many of these policies are being learned or adapted during deployment (Fig. 1). We leverage insights from the recent work by Paudel and Stein [13] whose *offline alt-policy replay* uses data collected by the robot from a navigation trial to simulate how another policy would have performed, yielding lower bounds on its performance that constrain and improve policy selection. As this approach does not consider that policies may change during deployment, the rolling estimates of the performance of each policy quickly diverge from their true expected performance as old performance estimates become stale. Instead, we leverage *offline alt-policy replay* to refresh old data and so compute up-to-date bounds on the would-be performance of each policy, allowing us to quickly select the best performing policy even as the robot's policies improve during deployment.

We deploy our approach in simulated maze-like environments where visual cues indicate promising routes to the unseen goal. Via our approach, our robot is deployed with six policies—a static pre-trained policy, a non-learned baseline policy, two policies being continually adapted via visual domain adaptation, and two policies being trained from scratch using deployment-time data—and quickly chooses the best performing ones among these when deployed. Notably, our approach correctly avoids the non-stationary policies early in deployment before they learn to understand the environment yet quickly switches to them once they improve, a capability that allows our approach to outperform all single-policy strategies and existing policy selection baselines.

## 2   Related Work

**Domain Adaptation**   Advances in visual domain adaptation [14, 15, 16, 17, 18, 19] have made it possible to adapt systems trained in one domain to a different domain during deployment. Visual domain adaptation approaches have been used to compensate for differences in the robot's visual observations between the training and deployment environments [20, 21, 22, 23, 24]. However, visual domain adaptation approaches, such as those that leverage CycleGAN [19], are not always suitable for improving performance, especially when domains are drastically dissimilar, requiring careful consideration for their uses on robotic systems where reliability is of critical importance.

**Deployment-Time Adaptation**   The idea of adapting robots during deployment has been explored widely in the robotics literature. Many approaches focus on short-horizon robot behavior such as adapting low-level motor controls or locomotion in diverse terrains [25, 26, 27, 28, 29, 30]. As such, they are not concerned with long-horizon behavior where the robot should consider the impacts of its immediate actions far into the future, the focus of this work.

**Policy Selection** can be thought of as model selection applied to choosing between robot behaviors [13]. In reinforcement learning, model selection [9, 10, 11, 12] is often treated as a generalized form of multi-armed bandit problem [31]. Using bandit algorithms [1, 2, 3]—which are often blackbox—for policy selection requires trade off between exploitation and exploration and hence the robot has to potentially go through multiple trials with poor behavior before a better policy can be identified. White-box policy selection [13] has shown recent promise in data efficiency, yet so far only consider stationary policy selection, ill-suited for policies that evolve during deployment.

## 3 Preliminaries

For robots to perform well across a variety of unfamiliar environments, they must have the ability to simultaneously apply multiple deployment-time learning and domain adaptation techniques in parallel, so that they may choose the best-performing of them when deployed. We model this scenario as an instance of *policy selection*. While the policies themselves are *non-stationary*—they change over time as the robot is deployed and collects data—we must first discuss the fundamentals of goal-directed navigation in partially-mapped environments and strategies for *stationary* (static) policy selection in this domain, needed to understand our approach.

**Goal-Directed Navigation in Partially-Mapped Environments** For each trial, our robot is placed in a partially-mapped environment and tasked to reach an unseen point-goal in minimum expected cost, measured in units of distance. Planning under uncertainty is often formulated as a partially observable Markov decision process (POMDP) [32]. As planning via the POMDP model directly is computationally intractable, many planning strategies in this domain rely on learning to anticipate what may lie in unseen space and thus inform good behavior. The robot's policy $\pi$ consumes a belief state $b_t$—including the robot pose $q_t$, the partial map $m_t$, and visual observations collected onboard the robot $o_t$—and returns a primitive action $a_t$ specifying the robot's behavior. For all policies in this work, images are used to inform learning, which makes predictions about unseen space to guide the robot more quickly to the unseen goal.

**Policy Selection over Multi-Trial Deployments** A single *deployment* consists of a sequence of trials, each a single traversal from start to goal in a previously-unseen map. We consider the scenario in which the robot is deployed with an ensemble of policies $\Pi = \{\pi_1, \pi_2, \cdots, \pi_N\}$, each defining a different behavior. The objective of policy selection is to determine, during a deployment, which of those policies performs best in the deployment-time environments:

$$\pi^* = \underset{\pi \in \Pi}{\operatorname{argmin}} \mathbb{E}[C(\pi)], \tag{1}$$

where $\mathbb{E}[C(\pi)]$ is the expected cost of policy $\pi$ in the deployment environment distribution. In practice, the expected cost is approximated by deploying the policy multiple times and averaging the performance: $\mathbb{E}[C(\pi)] \approx \bar{C}(\pi)$, the average cost of the trials in which policy $\pi$ was deployed.

### 3.1 Black-box Policy Selection via Multi-Armed Bandits

Selecting the policy that minimizes $\bar{C}(\pi)$ is unwise in general, since the randomness inherent in the deployments means that the best-performing policy may be incorrectly ruled out early on and never selected again to improve the estimate of $\mathbb{E}[C(\pi)]$ with no recourse to recover. Instead, the upper confidence bound (UCB) multi-armed bandit [1] balances exploitation with exploration, incentivizing selection of policies that have been chosen less often to improve the estimate of $\mathbb{E}[C(\pi)]$. For trial $k + 1$, UCB bandit selection specifies one choose policy $\pi^{(k+1)}$ according to

$$\pi^{(k+1)} = \underset{\pi \in \Pi}{\operatorname{argmin}} \left[ \bar{C}_k(\pi) - c\sqrt{\frac{\ln k}{n_k(\pi)}} \right], \tag{2}$$

where $\bar{C}_k(\pi)$ is the average cost over trials 1-through-$k$ in which policy $\pi$ was selected, $n_k(\pi)$ is the number of times policy $\pi$ was selected through trial $k$, and $c > 0$ is a parameter controlling the rate of exploration. In practice, each trial of goal-directed navigation is expensive and the resulting cost

samples have high variance. Thus, black-box policy selection approaches such as this are problematically slow to converge, making them impractical for policy selection in this domain despite their desirable guarantees on asymptotic sub-linear regret.

## 3.2 Data-Efficient Policy Selection via Offline Alt-Policy Replay

A central challenge of black-box model selection approaches is their data inefficiency, as each deployment yields only a single data point for a single policy. If it were instead possible to use the images and partial map collected during a trial to determine how well other policies could have performed—a bound on performance—we could use it to inform policy selection and so accelerate convergence to the best-performing policy. Paudel and Stein [13] present an approach they term *offline alt-policy replay* in which data collected during deployment of policy $\pi$ is used to simulate the would-be behavior of another policy $\pi'$ placed in the same environment.

Though we deploy a policy $\pi_{\text{DEPLOY}}$, we might want to know how well another policy $\pi_{\text{ALT}}$ could have done:

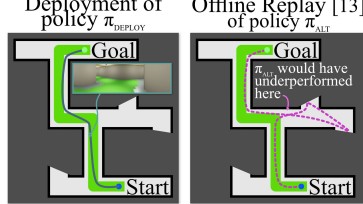

Using data from deployment of $\pi_{\text{DEPLOY}}$, offline alt-policy replay [13] computes a lower bound $\hat{C}^{\text{lb}}(\pi_{\text{ALT}})$ on the would-be cost of deploying another policy $\pi_{\text{ALT}}$.

Figure 2: Offline Replay Overview

Under this approach, deployment of policy $\pi$ during trial $k$ yields a *record* $\mathcal{Z}_k$ of all its poses and observations of the environment. This partial snapshot of the environment is used to simulate how another policy $\pi'$ would have acted, letting it navigate and reveal the environment and receive the image observations it needs to make predictions about unseen space that inform its behavior. Offline replay produces a lower bound on the performance of $\pi'$ had it instead been deployed: $C_k^{\text{lb}}(\pi') = \text{OFFLINEREPLAY}(\pi', \mathcal{Z}_k) \leq C_k(\pi')$. Thus, each trial yields information about the performance of *all policies* $\pi \in \Pi$. The average lower bound after $k$ trials, $\bar{C}_k^{\text{lb}}(\pi')$, constrains bandit-like policy selection:

$$\pi^{(\text{k+1})} = \operatorname*{argmin}_{\pi \in \Pi} \left[ \max \left( \bar{C}_k^{\text{lb}}(\pi), \bar{C}_k(\pi) - c\sqrt{\frac{\ln k}{n_k(\pi)}} \right) \right] \tag{3}$$

Offline alt-policy replay results in dramatically faster convergence to the best-performing policy while still preserving guarantees on asymptotic sub-linear regret. However, like the bandit algorithm before it, this approach presumes that the policies themselves are stationary, and so is not well-suited to select between policies being learned or adapted online, the focus of this work.

## 3.3 Learning over Subgoals Planning: Learning-Augmented Model-Based Planning

A key enabler of offline alt-policy reply is the requirement that planning be done via a policy amenable to counterfactual reasoning—i.e., *What would policy $\pi'$ have done if it had instead been placed in this situation?*—with which the behavior of policy $\pi'$ can be simulated. As such, our policies are based on the learning-informed model-based *learning over subgoals planning* (LSP) by Stein et al. [33], which satisfies this requirement.

Learning over subgoals planning (LSP) is a high-level planning framework for learning-informed model-based navigation in a partially-mapped environments. Under this abstraction, *subgoals* correspond to frontiers, each a boundary between free and unknown space; high-level

In learning over subgoals planning (LSP), actions correspond to navigating to frontiers:

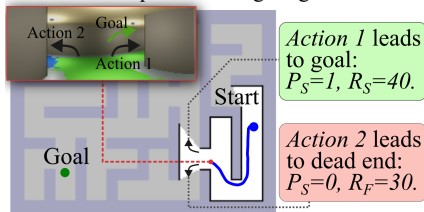

Learning estimates the likelihood ($P_S$) that the frontier-action leads to the goal and its associated costs ($R_S$ and $R_F$). Using these estimates, planning is done via Eq. (4).

Figure 3: Overview of learning over subgoals planning (LSP).

actions $a_t$ correspond to navigation to a subgoal and then exploration beyond to attempt to reach the unseen goal. The robot uses a learned model to make predictions about unseen space beyond each subgoal, information it uses to decide where to reveal next. The learned *subgoal property estimator*, a neural network $\mathcal{N}_\theta$ parameterized by $\theta$ and trained via supervised learning, consumes panoramic images in the vicinity of each subgoal to estimate the likelihood that each subgoal-action will suc-

cessfully reach the goal $P_{S,\theta}$ and the expected costs associated with success $R_{S,\theta}$ and failure $R_{F,\theta}$ to reach the goal in unseen space. Upon failing to reach the goal via a high-level action $a_t$, the robot must select another action $a \in \mathcal{A} \setminus a_t$. Planning is model-based, with the expected cost of a high-level action defined by a Bellman Equation:

$$Q_\theta(b_t, a_t) = D(b_t, a_t) + P_{S,\theta}(a_t) R_{S,\theta}(a_t) + (1 - P_{S,\theta}(a_t)) \left[ R_{F,\theta}(a_t) + \min_{a \in \mathcal{A}(b_t) \setminus a_t} Q_\theta(\tilde{b}'_t, a) \right] \quad (4)$$

where $b_t$ is the robot's belief, $D(b_t, a_t)$ is the travel cost to reach the subgoal $a_t$ via known space, and $\tilde{b}'_t$ is the approximate updated belief, which reflects that the robot has moved to the subgoal $a_t$. Planning seeks to find the action $a_t$ that minimizes expected cost: $\pi_\theta(b_t) = \mathrm{argmin}_a Q_\theta(b_t, a)$; the robot replans whenever the map is updated, proceeding until the goal is reached.

As our approach builds upon the insights of offline replay, all navigation policies in this work rely on the learning over subgoals planning abstraction. Planning for each is done via Eq. (4), and so policy selection in this context can be thought of as choosing the network model $\mathcal{N}_\theta$ whose predictions about unseen space $\{P_{S,\theta}, R_{S,\theta}, R_{F,\theta}\}$ result in the best performance when deployed.

# 4 Multi-Strategy Deployment-Time Learning and Adaptation

## 4.1 Problem Formulation: Non-stationary Policy Selection

We seek to achieve minimum-expected-cost performance for navigation in partially-revealed environments. Our robot is deployed with an ensemble of learning-informed policies—many of which are being learned or adapted after each trial—and seeks to pick the best performing strategy during deployment. We formulate this problem as an instance of *non-stationary policy selection*. A deployment is a sequence of $T$ trials, each a navigation from start to goal in a previously unseen map. As many of the policies are learned or adapted and so evolve over time, we add an additional subscript $k$ to denote the policy after trial $k$. Thus, before trial $k + 1$, the robot has access to $N$ policies $\Pi_k = \{\pi_{1,k}, \pi_{2,k}, \ldots, \pi_{N,k}\}$ and seeks to pick the one that minimizes expected cost via

$$\pi^{(k+1)} = \underset{\pi_{n,k} \in \Pi_k}{\mathrm{argmin}} \, \mathbb{E}[C(\pi_{n,k})] \, . \quad (5)$$

Though Eq. (5) resembles Eq. (1), computing the expected cost of policies in Eq. (5) is challenging because the policies themselves are continually being updated via learning or adaptation during deployment. The policy selection strategies discussed in Sec. 3 (Eq. (2) & (3)) use a rolling average to estimate $\mathbb{E}[C(\pi_{n,k})]$, which quickly diverges from the true estimate for policies that improve via deployment-time training or adaptation, reducing the robot's performance. Instead, if selection is to quickly and reliably converge to the best performing policy, there is a need for an approach that can compute accurate bounds on the performance of each policy $\pi_{n,k}$ even as they improve.

## 4.2 Approach: Selection over Non-Stationary Policies being Continuously Learned or Adapted During Deployment

We present an approach that performs data efficient policy selection over a set of policies that are continually learned or adapted during deployment. Our policy selection approach chooses policies based on the selection strategy of Eq. (3), yet rather than using *rolling averages* of the lower-bound costs $\bar{C}_k^{\mathrm{lb}}$ and the deployment costs $\bar{C}_k$ that fail to consider the evolving nature of the robot's policies, we instead rely upon OFFLINEREPLAY to refresh the estimates of the robot's performance.

**Computing Up-to-Date Bounds on Performance** Whenever one of the robot's policies is updated, via domain adaptation or learning with data it collects during deployment, its behavior may have changed. Thus, $C_k(\pi_{n,k}) \neq C_k(\pi_{n,k-1}) \neq \cdots \neq C_k(\pi_{n,1})$ in general and so using a rolling average of performance will result in poor selection performance and miss out on choosing policies that may have dramatically improved. Just as offline alt-policy replay can be used to determine the would-be performance of alternate policies after each trial, we can leverage this approach to revisit *old trials* to determine how well the robot's updated policies would have performed. We use the data

the robot collects until trial $k$ (the records $\{\mathcal{Z}_i\}_{i=1,\cdots,k}$) to replay how each of the robot's updated policies would have behaved in older trials, letting us get a much more accurate estimate of that policy's expected performance and performance bounds. For each updated policy $\pi_{n,k} \in \Pi_k$, the updated lower bound performance is computed by replaying its behavior for all trials:

$$\bar{C}_k^{\text{lb}}(\pi_{n,k}) \approx \frac{1}{k} \sum_{i=1}^{k} \text{OFFLINEREPLAY}(\pi_{n,k}, \mathcal{Z}_i) \tag{6}$$

The average cost of deployed policies $\bar{C}_k(\pi_{n,k})$ is recomputed similarly. We note that our problem setting envisions that robots will not be in constant operation and can perform much of this computation for training or re-evaluation while idle and waiting for its next navigation objective.

**Policy Selection using Updated Performance Estimates** We use the selection approach of Eq. (3) to perform policy selection, yet use our *updated* estimates of the performance of the robot's policies—lower-bound costs $\bar{C}_k^{\text{lb}}$ and the deployment costs $\bar{C}_k$ computed via Eq. (6)—in place of their rolling averages. After each trial $k$, the robot initiates a procedure to learn or adapt its non-stationary policies using the data it has so far collected $\{\mathcal{Z}_i\}_{i=1,\cdots,k}$. Before trial $k + 1$, it computes updated performance estimates for each of its updated policies $\pi_{n,k} \in \Pi_k$ via Eq. (6) and selects the policy to deploy via Eq. (3). Since data from new trials are continually added to record $\mathcal{Z}$ for training and offline replay, these policies will asymptotically converge to static policies after many trials, and we preserve bandit-like guarantees on sub-linear asymptotic regret.

## 4.3 Deploying an Ensemble of Policies

Facilitated by our approach, we deploy our robot with an ensemble of policies; some are unchanging (e.g., the robot's pre-trained policy) and others evolve and improve during deployment as more data is collected—e.g., those that rely on visual domain adaptation or are trained from scratch. Policies in the ensemble are chosen so as to capture potentially distinct scenarios during deployment. Each such policy depends on the learning over subgoals planning (LSP) approach discussed in Sec. 3.3, and so policy selection and domain adaptation in this context can be thought of as simultaneously improving and choosing between feed-forward models that make predictions about unseen space. Here, we discuss the different strategies employed by each policy our robot is deployed with.

**The Pre-Trained Learning-Informed Policy [Stationary] $\pi_{\text{PRETRAIN}}$** The robot is equipped with a static policy trained in advance of deployment, a learning over subgoals planning (LSP) policy well-suited to good performance in the training environments. If we were to know in advance that the deployment-time environments matched the training environments, we would expect this policy to perform well and so no policy selection would be necessary. However, this work considers the more general case wherein the deployment environments may differ non-trivially compared to those seen during training, leading to poor performance when relying on $\pi_{\text{PRETRAIN}}$.

**The Non-Learned Optimistic Policy [Stationary] $\pi_{\text{NONLEARNED}}$** We also deploy the robot with an *optimistic* planning strategy: a common non-learned strategy in which unseen space is assumed to be unoccupied. The non-learned optimistic policy can be thought of as a special case of LSP, in which $P_S \equiv 1$ and $R_S, R_F \equiv 0$, so that the robot simply selects the shortest path through the partial map to the unseen goal, replanning when necessary as unseen space is revealed. This strategy of optimism under uncertainty is unlikely to outperform $\pi_{\text{PRETRAIN}}$ when the deployment environments match those seen during training. However, when deployed in an unfamiliar environment that differs significantly from the training environment, the learned model underpinning $\pi_{\text{PRETRAIN}}$ may be misled by visual features it cannot properly understand, making $\pi_{\text{NONLEARNED}}$ a reasonable backup strategy.

**Visual Domain Adaptation via CycleGAN [Non-Stationary] $\pi_{\text{CYCLEGAN}}$** Our robot will use images it collects during deployment to perform *visual domain adaptation* via the CycleGAN algorithm [19, 21]. Using unlabeled images from the training and deployment environments, CycleGAN learns a mapping from one to the other, as shown in Fig. 4. This mapping is used as a preprocessing step: deployment-time images are made to look like training-time images and then fed to the robot's pre-trained policy $\pi_{\text{PRETRAIN}}$. The resulting policy $\pi_{\text{CYCLEGAN}}$ can thus compensate for visual

dissimilarities between training and deployment, and so can improve performance when such changes are what limits the robot's behavior. However, visual domain adaptation *cannot* perform well in all deployment environments, something policy selection must judge during deployment. For our experiments, we train two models, one for 50 epochs $\pi_{\text{CYCLEGAN}_{50}}$ and one for 100 epochs $\pi_{\text{CYCLEGAN}_{100}}$, and allow policy selection to choose between them. Additional details of the CycleGAN approach and training can be found in the Appendix.

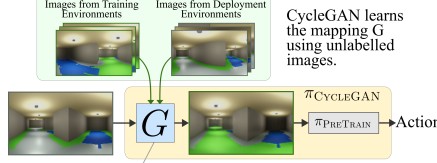

Figure 4: Adaptation of a pre-trained policy using visual domain adaptation.

**Training from Scratch during Deployment [Non-Stationary]** $\pi_{\text{SCRATCH}}$ Using data it collects during deployment, the robot can additionally train an LSP-style policy *from scratch* while deployed. For LSP, the robot's learned model estimates for each subgoal (associated with a boundary between free and unseen space that may lead to the unseen goal) the likelihood $P_S$ that the subgoal successfully leads to the goal and the expected costs associated with reaching the goal $R_S$ or needing to explore and turn back $R_F$. As the robot navigates, it can label image data using the partial map to train an LSP model: routes the robot discovered to reach the goal correspond to $P_S = 1$ and dead-ends to $P_S = 0$. Ambiguous routes—potential routes to the goal the robot did not explore during deployment—can also be included in the training data under either an *optimistic* prior, i.e., where all ambiguous routes are labeled with $P_S = 1$, or a *conservative* prior, i.e., an assumption that unexplored space is simply-connected, so that $P_S = 0$. During deployment, we train one such policy for each of the optimistic prior $\pi_{\text{SCRATCHOPT}}$ and conservative prior $\pi_{\text{SCRATCHCON}}$. While these policies will be invariably slow-to-converge towards good performance, they have the potential to successfully improve performance even in environments where visual domain adaptation is not well-suited. We use 5-fold cross validation to determine an unbiased estimate of the updated lower bound on performance and deploy a policy trained on all trials.

## 5  Experimental Results

We demonstrate the effectiveness of our approach in simulated maze-like environments, procedurally generated so that each navigation trial sees a unique map. Each deployment consists of 50 navigation trials, each a traversal from start to goal through a previously-unseen map. As described in Sec. 4.3, the robot has an ensemble of six policies to choose from during deployment: $\Pi = \{\pi_{\text{NONLEARNED}}, \pi_{\text{PRETRAIN}}, \pi_{\text{CYCLEGAN}_{50}}, \pi_{\text{CYCLEGAN}_{100}}, \pi_{\text{SCRATCHOPT}}, \pi_{\text{SCRATCHCON}}\}$; to save computation, the non-stationary policies are updated via learning or adaptation only every 10 trials. In addition to our approach for non-stationary policy selection NONSTATIONARYREPLAY, we include results with UCBBANDIT and ROLLINGREPLAY [13] policy selection, which performs selection via Eq. (3). See the Appendix for more details on the choice of these baselines.

We deploy in three maze variants, each with differing (often conflicting) visual cues that signal routes to the unseen goal. Results (Fig. 5) show average performance and cumulative regret of each selection strategy over time. We additionally show (Fig. 5 bottom) our NONSTATIONARYREPLAY's estimate of the cost lower bound $\bar{C}^{\text{lb}}$ over time, lending insight into its selection process.

**Deployment in Maze-Green:** *A green path signals routes to the goal, the floor is gray, and a blue path leads to dead-ends.* The stationary $\pi_{\text{PRETRAIN}}$ is trained offline in held-out maps from this environment and so is the best. While the UCBBANDIT converges slowly, both NONSTATIONARYREPLAY and ROLLINGREPLAY expectedly select $\pi_{\text{PRETRAIN}}$ quickly, achieving near-optimal performance.

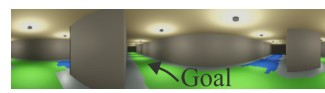

**Deployment in Maze-Gray:** *A gray path signals routes to goal, the floor is green, and the blue path leads to dead ends.* Misled by the green flooring, $\pi_{\text{PRETRAIN}}$ performs poorly and so both NONSTATIONARYREPLAY and ROLLINGREPLAY, initially select the non-learned $\pi_{\text{NONLEARNED}}$. However, only our approach considers that policies may evolve during deployment. After 10 trials, the

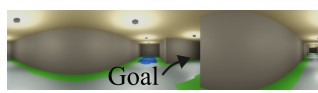

| | Deployment Environment | UCB BANDIT | ROLLING REPLAY [13] | NONSTATIONARY REPLAY (ours) | Our Improvement vs. UCB | vs. ROLLING |
|---|---|---|---|---|---|---|
| **Avg. Cost** | Maze-Green | 187.9 | **161.6** | **161.6** | 14.0% | 0.0% |
| | Maze-Gray | 183.7 | 189.3 | **165.0** | 10.2% | 12.8% |
| | Maze-Blue | 243.5 | 202.3 | **196.5** | 19.3% | 2.9% |
| **Cumul. Regret** | Maze-Green | 1375.2 | **61.2** | **61.2** | 95.6% | 0.0% |
| | Maze-Gray | 1848.6 | 2131.2 | **916.2** | 50.4% | 57.0% |
| | Maze-Blue | 2955.6 | 898.2 | **606.6** | 79.5% | 32.5% |

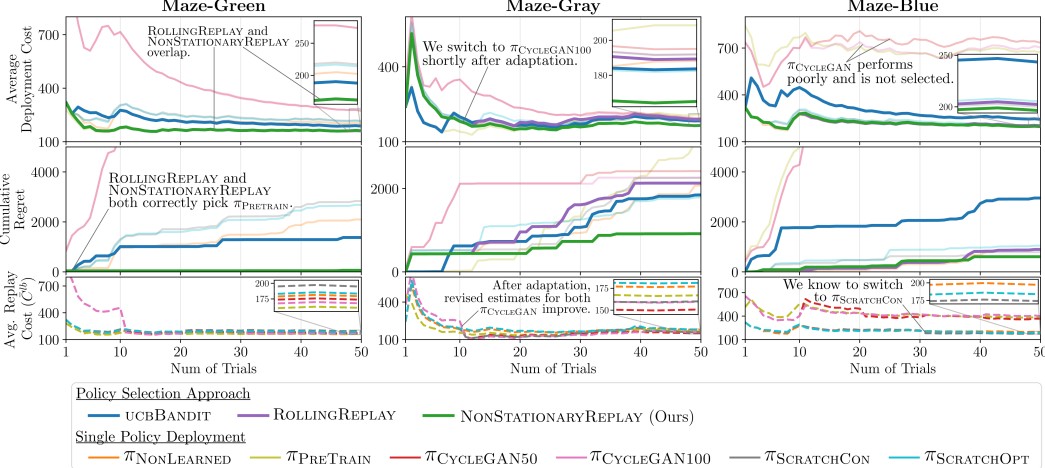

Figure 5: Our NONSTATIONARYREPLAY policy selection approach outperforms both UCBBANDIT and ROLLINGREPLAY [13] baselines in both Average Navigation Cost and Cumulative Regret.

$\pi_{\text{CycleGAN}}$ policies improve via visual domain adaptation and our approach quickly switches to them, achieving 57% lower cumulative regret compared to ROLLINGREPLAY. Notably, selecting $\pi_{\text{CycleGAN}}$ before it improves would *reduce* performance; thus, our NONSTATIONARYREPLAY *outperforms all single-policy selection strategies*.

**Deployment in Maze-Blue:** *A blue path on the ground signals routes to goal, the floor is gray, and the green path leads to dead ends.* $\pi_{\text{CycleGAN}}$ cannot resolve the visual dissimilarities between

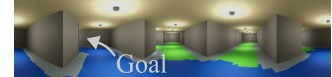

training and deployment even after many trials, and so the policies trained from scratch $\pi_{\text{Scratch}}$ are the most promising approaches despite their slow convergence. After 30 trials, $\pi_{\text{ScratchCon}}$ improves sufficiently and our NONSTATIONARYREPLAY selects it, resulting in 32% lower regret compared to ROLLINGREPLAY, a number that would continue to grow with further trials.

# 6 Conclusion, Limitations, and Future Work

We present an approach to monitor and quickly select between learning-informed navigation policies, many of which are being continuously learned or adapted during deployment. Our approach facilitates deploying multiple-such strategies for learning and visual domain adaptation in parallel, allowing our robot to choose the best among them over time as demonstrated in simulated visual maze-like environments in which we outperform state-of-the-art selection strategies by a considerable margin. In future, we aim apply our approach to other domains like multi-robot planning and task planing under uncertainty since these problems also show promise for formulation in a way that allows counterfactual reasoning despite uncertainty and so hold promise for applying our approach.

**Limitations** Although our policy selection approach is not particularly computationally intensive on its own, each strategy for learning/adaptation relies on training a deep neural network, and so requires considerable computation as the number of non-stationary policies grows (see the Appendix for further discussion). Moreover, our policy selection method relies on policies amenable to counterfactual reasoning—of which learning over subgoals planning (LSP) is one—and so may not be directly applicable for selection between model-free navigation methods, limiting its broad utility.

**Acknowledgments**

This material is based upon work supported by the National Science Foundation (NSF) under Grant No. 2232733. This work was done at Robotic Anticipatory Intelligence & Learning (RAIL) Group and RobotiXX Laboratory at George Mason University. RobotiXX research is supported by National Science Foundation (NSF, 2350352), Army Research Office (ARO, W911NF2220242, W911NF2320004, W911NF2420027), US Air Forces Central (AFCENT), Google DeepMind (GDM), Clearpath Robotics, and Raytheon Technologies (RTX).

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

# Multi-Strategy Deployment-Time Learning and Adaptation for Navigation under Uncertainty: Appendix

## A    CycleGAN Implementation and Training

We use the official PyTorch implementation of CycleGAN provided by Zhu et al. [19] on GitHub[1]. As mentioned in Sec. 4.3, we train two models: one for 50 epoch and another for 100 epochs. We use the default parameters for training except that we use a batch size of 8 and no learning rate scheduling is done. We use learning rate of 0.0002 with Adam optimizer. The images are of size $512 \times 128$ and we perform default resizing and cropping as a preprocessing step. 1261 images from 10 distinct maps in Maze-Green (where $\pi_{\text{PRETRAIN}}$ is trained) are set aside as target domain images, and we sample 1300 images from the source domain (either Maze-Gray or Maze-Blue) collected during deployment for training the CycleGAN model. Using only 1300 images from deployment-time environments additionally helps to get an unbiased estimate of performance when replaying CycleGAN-adapted policy $\pi_{\text{CYCLEGAN}}$ on older trials from which the images were collected.

## B    Learning over Subgoals Planning: Subgoal Property Estimator Network Implementation and Training

As discussed in Sec. 3.3, all learning-informed policies rely on the learning over subgoals planning (LSP) abstraction for planning, a model-based approach that relies on a learned model to estimate *subgoal properties*: statistics of unseen space associated with each of the robots temporally-extended high-level actions to explore unseen space. The subgoal property estimator network $\mathcal{N}_\theta$ corresponding to $\pi_{\text{PRETRAIN}}$ is trained with data-collected in 500 distinct maps in Maze-Green where the robot navigates using the $\pi_{\text{NONLEARNED}}$ policy. Data labelling procedure is similar to the one described for $\pi_{\text{SCRATCH}}$ (Sec. 4.3) except that at training time the underlying map is known and so can be used to generate ground truth labels for subgoal properties $P_S, R_S$ and $R_F$ corresponding to all subgoals.

The subgoal property estimator network $\mathcal{N}_\theta$ is trained via supervised learning using the data collected during an offline training phase. Our neural network architecture and training procedure resemble that of Paudel and Stein [13]. The network takes as input a $512 \times 128$ panoramic image centered on a subgoal, relative distance to the subgoal and relative distance to the goal. The image is encoded by passing through 4 convolutional layers and then concatenated with features corresponding to relative distances to subgoal and goal after which the concatenated features are passed through 9 convolutional layers and finally 5 fully connected layers to output 3 subgoal properties $P_S, R_S$ and $R_F$. We use a learning rate of 0.002 with a decay factor of 0.5 every epoch and train for 8 epochs with Adam optimizer. We use cross-entropy loss for learning logits associated with $P_S$ and L2 loss for learning $R_S$ and $R_F$. The deployment-time training of subgoal property estimators for $\pi_{\text{SCRATCH}}$ follows a similar architecture and procedure.

## C    Offline Alt-Policy Replay Details

As discussed in Sec. 3.2, we use offline alt-policy replay to replay the behavior of a policy without deploying it. To replay the behavior of a policy $\pi'$, we leverage the record $\mathcal{Z}_k$ collected during trial $k$ under a deployed policy $\pi$. At every time step during offline alt-policy replay, the robot leverages the final partial map $m_{\text{final}}$ observed in trial $k$ to simulate the laser scan and updates its observed map as the robot moves. The frontiers—boundaries between free and unknown space—revealed in the observed map corresponds to the subgoal-actions that the robot can take to explore the region. To get the robot-view panoramic image corresponding to a subgoal-action, we retrieve

---

[1] https://github.com/junyanz/pytorch-CycleGAN-and-pix2pix

existing image in record $\mathcal{Z}_k$ that is closest to and in line of sight to the subgoal and recenter it at the subgoal. This image is used to estimate the subgoal properties $P_S, R_S$ and $R_F$ using a neural network corresponding to $\pi'$ which is then used to compute the next high-level action using Eq. (4). The robot then simulates the low-level motion primitive to move towards the selected subgoal and this process is repeated. At any point during simulated navigation, if the robot attempts to enter a region that is unknown in the final partial map $m_{\mathrm{final}}$ via a frontier, we mask that frontier and force the robot to pick a different subgoal-action. The net distance travelled to reach the goal via this procedure is the replay cost of policy $\pi'$.

## D   Cross Validation for Reevaluating Older Trials

As mentioned in Sec. 4.3, we use 5-fold cross validation to get an unbiased estimate of the performance of updated policies. Since our policies trained during deployment from scratch ($\pi_{\mathrm{SCRATCH}}$) are based on the same data in record $\mathcal{Z}$ that is also used for offline alt-policy replay to reevaluate older trials after updating the policies, such cross-validation approach overcomes the risk of overestimation of performance during replay due to data leak. With 5-fold cross validation, 5 policies are trained, each on the data from four-fifth of older trials, and replayed on the remaining one-fifth of the trials to get the revised performance estimates for all older trials. Finally, a new policy is trained on data from all previous trials and made available for the robot to choose from in the next trial.

## E   Ablation Studies

We study the effect of removing the most effective learning/adaptation strategies from Maze-Gray and Maze-Green and see how the performance varies. Specifically, we remove CycleGAN-based policies ($\pi_{\mathrm{CYCLEGAN}_{50}}$ and $\pi_{\mathrm{CYCLEGAN}_{100}}$) from Maze-Gray and policies trained from scratch ($\pi_{\mathrm{SCRATCHCON}}$ and $\pi_{\mathrm{SCRATCHOPT}}$) from Maze-Blue. The results are shown in Table 1. We observe that in both environments, removing the corresponding best learning/adaptation strategies leads our NON-STATIONARYREPLAY approach still outperforming or on-par with baselines. We also observe that the average navigation cost for our NONSTATIONARYREPLAY approach increases when the best learning/adaptation strategies are removed compared to those in Fig. 5 where these strategies were included.

Table 1: Results of ablation with removing best learning/adaptation strategies corresponding to Maze-Gray and Maze-Blue

|  | Environment/ Ablation | UCB BANDIT | ROLLING REPLAY [13] | NONSTATIONARY REPLAY (ours) | Our Improvement vs. UCB | vs. ROLLING |
|---|---|---|---|---|---|---|
| **Average Cost** | Maze-Gray (w/o $\pi_{\mathrm{CYCLEGAN}}$) | 190.8 | 197.1 | **176.3** | 7.6% | 10.56% |
|  | Maze-Blue (w/o $\pi_{\mathrm{SCRATCH}}$) | 234.6 | **202.3** | **202.3** | 13.8% | 0.0% |
| **Cumulative Regret** | Maze-Gray (w/o $\pi_{\mathrm{CYCLEGAN}}$) | 1917.0 | 2232.0 | **1191.6** | 37.8% | 46.6% |
|  | Maze-Blue (w/o $\pi_{\mathrm{SCRATCH}}$) | 2053.8 | **441.0** | **441.0** | 78.5% | 0.0% |

## F   Additional Discussion of Trends in Results

**Adaptation of $\pi_{\mathrm{CYCLEGAN}}$ in Maze-Gray and Maze-Blue**   As is mentioned in Sec. 5, Maze-Blue has blue path signaling routes to goal and green path leading to dead ends, and is deliberately designed to be indistinguishable from Maze-Green based only on visual observations in absence of the

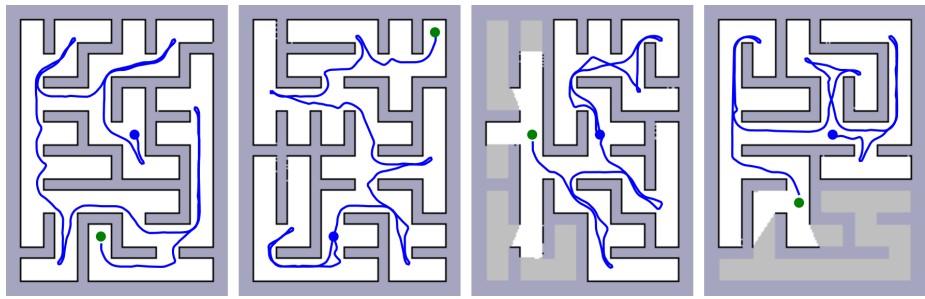

Figure 6: Sample Trajectories for $\pi_{\text{PreTrain}}$ in Maze-Blue

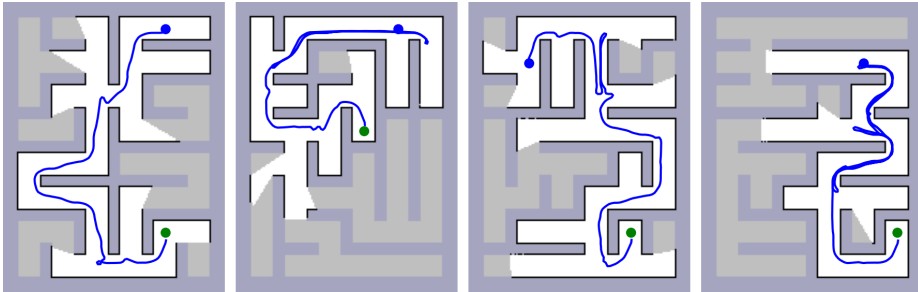

Figure 7: Sample Trajectories for $\pi_{\text{PreTrain}}$ in Maze-Gray

information about goal location. To clarify, CycleGAN does learn a reasonable visual mapping between the two environments: it learns an approximate identity mapping (see images corresponding to Maze-Blue in Fig. 8) from the visual observations from each. However, we note that this mapping is not helpful for resolving the best path to the goal since the robot is still drawn to follow the green path, which no longer leads in the direction of the unseen goal. Hence, the performance of $\pi_{\text{CycleGAN}}$ is poor in Maze-Blue and say that "visual domain adaptation fails" to properly adapt. On the other hand, in Maze-Gray, the color of the floor and goal-routes are swapped compared to Maze-Green. CycleGAN learns a mapping between the two that "swaps back" the colors of floor and path, a mapping that then allows the robot to correctly identify the best path to the goal, resulting in improved performance of $\pi_{\text{CycleGAN}}$ in Maze-Gray.

**Performance Differences of $\pi_{\text{PreTrain}}$ in Maze-Blue and Maze-Gray**   The $\pi_{\text{PreTrain}}$ policy trained in Maze-Green would have learned to follow the green path to find the goal and avoid the blue path to dead ends. As such, deploying it in Maze-Blue where green path leads to dead ends and blue path leads to goal (as illustrated in the respective environment's figures in Sec. 5) would often mislead $\pi_{\text{PreTrain}}$ to navigate towards the dead ends increasing the cost to find the goal (see example trajectories in these environments as shown in Fig. 6 and 7). In Maze-Gray, this phenomenon is less severe since the dead ends are still blue and are often avoided by $\pi_{\text{PreTrain}}$.

**Performance Similarities of $\pi_{\text{PreTrain}}$ and $\pi_{\text{CycleGAN}}$ in Maze-Blue**   As discussed in the aforementioned paragraph, the visual observations from Maze-Blue and Maze-Green are similar in absence of the knowledge about goal. This causes CycleGAN generator in $\pi_{\text{CycleGAN}}$ to effectively learn an identity mapping (see images for Maze-Blue in Fig. 8) and therefore $\pi_{\text{CycleGAN}}$ policies are effectively equivalent to $\pi_{\text{PreTrain}}$ (see Fig. 4 on how $\pi_{\text{PreTrain}}$ is used by $\pi_{\text{CycleGAN}}$) resulting in very similar performances over trials in Fig 5.

## G    Sample Images from Deployment-Time Visual Domain Adaptation

Image samples transformed from deployment-time environments (Maze-Gray or Maze-Blue) to the training-time environment (Maze-Green) are shown in Fig. 8. The CycleGAN models trained after 40th trial are used to generate the images.

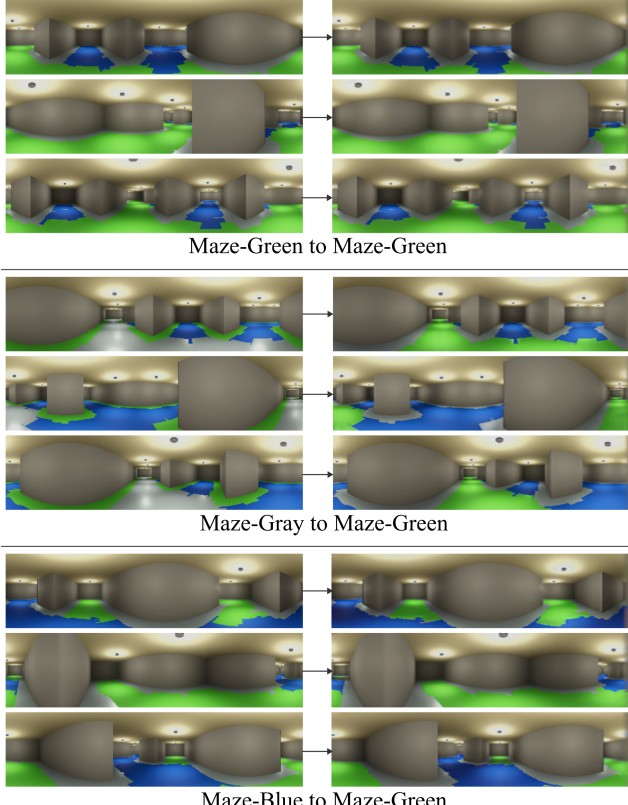

Maze-Green to Maze-Green

Maze-Gray to Maze-Green

Maze-Blue to Maze-Green

Figure 8: Images transformed from deployment-time environments (input) to look like training-time environments (output) with CycleGAN-based visual domain adaptation.

## H   Additional Related Work

**Runtime Monitoring**   Runtime monitoring approaches aim to evaluate the reliability of learned models used to guide the robot's behavior during deployment [4, 5, 6, 7, 8] and decide whether a fallback policy should instead be used. However, these approaches focus primarily on evaluating certain subsystems (e.g., the perception system) [5, 8] or scrutinizing a learned model's raw output instead of the robot's task performance [6, 7], thus limiting their applicability for evaluating the robot's long-horizon behavior for navigation in partially-mapped environments.

**Adaption with Meta-Reinforcement Learning**   Many problems in robotics have leveraged meta-reinforcement learning (meta-RL) approaches for learning robot policies that can be adapted to new tasks or scenarios [30, 34, 35, 36, 37]. While meta-RL is most effective when provided access during training to a distribution over environments to which the deployment-time environments may belong, our work is geared more towards reliably identifying and deploying existing general-purpose online training and domain adaptation tools meant to handle systematic differences between training and deployment-time environments. As such, our approach allows for integrating a wide range of existing learning/adaptation strategies as demonstrated in our experiments with a mix of different policies—something that meta-RL is not geared towards. Additionally, our approach makes no assumptions about similarities between training and deployment environments while still benefiting from asymptotic performance guarantees.

## I   Choice of Baselines

We compare our NONSTATIONARYREPLAY approach with UCBBANDIT and ROLLINGREPLAY baselines both of which assume that the policies are stationary. While it would have been more suitable to

compare our approach with non-stationary policy selection approaches, existing potential baselines in this area either assume policies are stationary or leverage knowledge we do not have access to in the context of robot navigation, e.g. underlying distribution of a policy's navigation cost. As such, ours is the first to propose such non-stationary policy selection algorithm for robot navigation under uncertainty, and hence our baselines only include the most suitable stationary policy selection approaches from the literature.

## J    Scalability and Computational Limitations

Our approach scales linearly in terms of both the number of policies and the number of trials: the addition of a new trial requires that offline alt-policy be run for the policies that were not deployed and, as policies are retrained, each old trial is re-evaluated. Such replay isn't computationally intensive—replaying a policy in a single typical trial takes roughly 20 seconds—and hence scales well over larger number of trials and policies. Additionally, our policy selection approach is not particularly computationally demanding on its own, but that the underlying strategies for training or adapting policies online (e.g., via CycleGAN) can be expensive, costs that are not specific to our policy selection approach, and so should not be seen as a limitation specific to our policy selection approach.

In real-world settings, the computation associated with training policies and selecting between them will not preclude using policy selection via our approach; specifically, our problem setting envisions that robots will not be in constant operation and can perform much of this computation and policy selection while idle and waiting for its next navigation objective. Thus, such computation associated with our approach would not necessarily impede navigation performance. For very lengthy deployments spanning thousands of trials, indeed our approach might run into practical limitations since determining an accurate estimate of the robot's performance requires replaying *all* previous trials. In future work, one could consider only selectively replaying specific trials or randomly sampling a subset of trials to replay to save on computation while preserving our asymptotic performance guarantees.

## K    Compute Platform and Execution Time

We ran our experiments on Intel i9 CPU with NVIDIA RTX A6000 GPU. As mentioned earlier, the most computational intensive tasks in our experiments are training the CycleGAN and the neural networks. While deploying the robot with our approach, training a policy from scratch (8 epochs using data from 50 maps) would take around 5 minutes, and training a CycleGAN (100 epochs with about 1300 images in each domain) would take around 3 hours. By contrast, performing offline alt-policy replay for a single policy on a single typical trial takes about 20 seconds.

