# OpenReview forum: "Multi-Strategy Deployment-Time Learning and Adaptation for Navigation under Uncertainty"
_robot-learning.org/CoRL/2024/Conference — CoRL 2024_

### Official Review · Reviewer_APKz · 2024-07-19
**Interesting setting, but with limited applicability**

**Originality:** 3
**Technical Quality:** 3
**Clarity Of Presentation:** 4
**Potential Impact:** 2
**Recommendation:** 2
**Confidence:** 4

**Review:**

## Strengths

The paper is clearly and simply presented, and is generally easy to read and follow along.

The setting presented is of interest to robotic practitioners, and one that could easily arise in many real-world applications.

The algorithm presented is in itself novel, and presents a viable solution to a challenging problem. The fact that it comes with an inherited regret bound is a great bonus.

I found the insights and observations in the paper enlightening. In particular, the observation about rolling average estimators not keeping up with adapting policies is insightful, as well as the characterization of environments in which domain adaptation policies may not be able to solve the task.

## Weaknesses

### Comparisons to other Approaches

At the start of the preliminaries section:

“For robots to perform well across a variety of unfamiliar environments, they must have the ability to simultaneously apply multiple deployment-time learning and domain adaptation techniques in parallel… “

This statement focuses on the proposed solution to the problem of performing in unfamiliar environments, but ignores many other available approaches. The first approach that comes to mind, especially when reading the “Policy Selection over Multi-Trial Deployments” setting, is Meta-RL, where a parameterized policy class is used to select a good test-time policy based on exploration and updating belief.

While meta-RL (and other similar settings) may not be directly applicable to the setting proposed in this paper without any changes, it would be beneficial to see fair comparisons to other approaches on the same environment to evaluate the strength of the proposed approach.

On a related note, lines 280-281 state: “Both assume that policies are stationary, and so are suitable baselines for comparison.”

Why does that make them suitable? Wouldn’t we want to compare to methods that do not assume policies are stationary?

### Computational Cost

While mentioned to some extent in the limitations section, I find the computational cost of the proposed approach to be the main weakness of the paper.

The computational cost of recomputing policy performance for all previous trials grows linearly with each trial. It seems like even for a low number of trials, this is very costly, considering the five-fold cross validation approach. Doesn’t it become infeasible at a certain number of trials, thus constraining this approach to problems with a short time horizon?

This is on top of the training time required for adaptive policies; for instance, the CycleGAN 50 and CycleGAN 100 policies are trained for 50 or 100 iterations respectively every 10 trials. How long does this training step take?

While this computational cost may still be feasible when running a robot in simulation, one may wonder if this approach can ever be applied to real-world robots, which usually have to operate with low compute budgets at near-real-time speeds.

### Limited Scope

As mentioned in the limitations section, the scope of the paper is limited to navigation problems, as it relies on the LSP framework for reevaluation of updated policies on past experience. This is due to the required property of counterfactual evaluation. It seems like in the general case, this would very quickly run into the problem of evaluation based on offline data, which is common in offline RL settings.

Have the authors considered ways to adapt their approach to more general domains? If this approach cannot be adapted to more general settings, its limited applicability may render it an incremental improvement for a specific robotic application.

**Quality Of The Limitations Section:**

3

**Questions For Rebuttal:**

See Review section above for questions and concerns.

**Robotics Focus:**

2

**Summary Of Paper:**

This paper presents a novel method for online selection of policies as an adaptation method to new visual navigation environments. The proposed method assumes access to an ensemble of policies, and takes into account their non-stationarity when selecting which policy to use. The selection algorithm reevaluates current policies on previous experience, relying on the counterfactual property of the LSP framework. Based on a previous approach, the algorithm proposed in this paper inherits lower bounds on cumulative regret.

**Summary Of Recommendation:**

While this paper proposes a valid approach to solving an interesting problem, it seems like its scope is limited, and it is hard to see how it can be applicable to real-world robotic applications beyond a specific use-case. Therefore, I recommend rejecting the paper.

---

### Official Review · Reviewer_rHoK · 2024-07-21
**Review of Submission 258**

**Originality:** 4
**Technical Quality:** 4
**Clarity Of Presentation:** 5
**Potential Impact:** 3
**Recommendation:** 3
**Confidence:** 4

**Review:**

## Strength:
1. The paper is very well written with thorough and intuitive explanations of all key concepts.
2. The problem presented is very relevant to robotics research, and the results presented in the experimental section are promising.

## Weakness:
1. The technical contribution of this paper is marginal since the key parts that address the core part of the proposed problem are based on existing approaches: LSP [27] enables policies that allow counterfactuals, and offline alt-policy replay [13] enables the calculation of would-be performances of non-stationary policies in the ensemble via learning. But I still believe there’s merit in extending previous work to address very challenging existing problems.
2. The authors missed the opportunity to demonstrate the approach in a real-world setting (especially since [27] was demonstrated on hardware). Another related concern is practicality in real-world settings since the proposed approach is very computationally heavy (addressed in the limitations section). Additionally, some trends in the results are not addressed (see questions 3 and 4).

**Quality Of The Limitations Section:**

3

**Questions For Rebuttal:**

1. What are the limits on the size of the ensemble of policies? Is the scalability only related to computation resources available? If not then would having more policies in the ensemble potentially help performance online?
2. Are there any insights into why CycleGAN adapts to Maze-Gray but not Maze-Blue?
3. It seems like the pretrained policy only exhibits a high cost in Maze-Blue but not in Maze-Gray from Figure 5 but in the main text (Sec. 5) it was mentioned that pretrained also performs poorly in Maze-Gray. Shouldn’t the performance be similar in both of these scenarios?
4. Related to Question 3, it seems like the performance of pretrained and CycleGAN policies are somewhat mirroring each other. Are there any explanations for this phenomenon?

**Robotics Focus:**

3

**Summary Of Paper:**

This paper presents a method for online policy selection from an ensemble of evolving, nonstationary policies being updated. The proposed method leverages offline alt-policy replay to compute would-be performances of policies, and perform selection based on these metrics.

**Summary Of Recommendation:**

Although the core components of the proposed approach are based on prior work, the combination of them still address a very challenging problem in planning, therefore I recommend weak accept. After rebuttal, the authors answered my questions on the results, and some ratings are updated accordingly.

---

### Official Review · Reviewer_UHRD · 2024-07-22
**Review of Submission 258 by Reviewer UHRD**

**Originality:** 4
**Technical Quality:** 4
**Clarity Of Presentation:** 4
**Potential Impact:** 3
**Recommendation:** 4
**Confidence:** 4

**Review:**

Strengths

- The overall idea of monitoring multiple potentially non-stationary strategies in parallel and quickly selecting the best one is interesting and potentially useful for robotic problems beyond navigation in unfamiliar environments.
- The paper is clear and well written, with a clear emphasis on what's already done and what's the contribution of this work throughout the paper.
- The approach preserves guarantees on asymptotic sub-linear regret while handling non-stationary policies.

Weaknesses:

- While I am satisfied with the experiments in the maze environment in terms of demonstrating the potential of proposed approach, one could argue that these may not represent the results you would expect in complex real-world scenarios. Performing experiments in more diverse test cases in simulators widely used in indoor navigation research [1] would have highlighted the practical utility of the work.

- There are no experiments isolating the contributions of individual components (e.g., CycleGAN vs. training from scratch for adaptation), making it difficult to understand how much the overall results vary depending on what you have in the set of candidate policies.

Nits:

- While acknowledged in the limitations, I think the paper glosses over the computational burden of this approach, especially in a real-robot setting.
- There are several recent works that focus on online and few-shot adaptation to handle deployment gaps in robotics (e.g, [2], [3], [4]) that should be acknowledged while talking about deployment-time learning and adaptation which is certainly not a new idea.
- It'll be nice to have some whitespace between the text and the illustrations in the figures - it looks a little too crowded right now.

[1] Habitat 3.0: A Co-Habitat for Humans, Avatars and Robots
[2] RMA: Rapid Motor Adaptation for Legged Robots
[3] Few-shot Adaptation for Manipulating Granular Materials Under Domain Shift
[4] Adapt On-The-Go: Behavior Modulation for Single Life Robot Deployment

**Quality Of The Limitations Section:**

3

**Questions For Rebuttal:**

- The approach is heavily dependent on the Learning over Subgoals Planning framework as offline alt-policy reply requires planning be done via a policy amenable to counterfactual reasoning. Could you elaborate on why this would mean a hard requirement of LSP?
- How realistic is a multi-trial per deployment assumption for a real world setting?
- How scalable is this approach, both in terms of number of policies in the pool and longer deployments with hundreds or thousands of trials?

**Robotics Focus:**

2

**Summary Of Paper:**

This paper presents an approach for efficient point-goal navigation in unfamiliar partially observable environments. The key idea is to deploy multiple strategies for deployment-time learning and visual domain adaptation in parallel, and then quickly select the best-performing strategy. A single deployment here is defined as a sequence of trials. The authors formulate this as a non-stationary policy selection problem and propose a novel method to estimate up-to-date performance bounds using offline alt-policy replay for policies that are continuously evolving during deployment . The approach is evaluated in simulated maze-like environments and shows promising results compared to baselines.

**Summary Of Recommendation:**

I think this paper explores an interesting research problem that will become more and more common as we start to have multiple competing policies for the same task in robotics. While the experiments of the paper are in simple simulated environments, I think they are sufficient to validate the main contributions of the work.

---

### Author Rebuttal · Authors · 2024-08-13

We would like to thank all the reviewers and the area chair for their thoughtful feedback on our work. We are glad that the reviewers found our paper interesting, well-presented and insightful. We are also excited by the fact that the reviewers believe our work to be relevant and impactful to the robotics community.

We have responded to the concerns of reviewers individually and have also included the necessary changes in the updated manuscript (highlighted in blue). With individual responses and the updated manuscript (see attached zip file), we believe that we have addressed all the concerns from the reviewers. We include a high-level summary of our comments in reply to the Meta-Review and provide more in-depth discussion and details in response to the individual reviewers’ comments.

---

### Decision · Program_Chairs · 2024-09-04

**Decision:**

Accept

**Comment:**

This paper received 3 fairly confident reviews. The manuscript presents an approach for efficient point-goal navigation in unfamiliar partially observable environments. The key idea is to deploy multiple strategies for deployment-time learning and visual domain adaptation in parallel, and then quickly select the best-performing strategy.

To address in rebuttal:
- Add results that isolate the contributions of individual components (ie, an ablation)
- Address scalability of approach wrt number of trials and number of policies; computational limitations
- Further address the nature (interpretation) of the results, as pointed out by reviewers.

Post-rebuttal:
The authors address many of the concerns raised raised by the reviewers; some doubts remain about the applicability of this approach to real-world robotic scenarios (the experiments of the paper are in simulated environments). Overall, the paper explores an interesting research problem in cases where multiple competing policies exist for the same task).